



# Consistency of total column ozone measurements between the Brewer and Dobson spectroradiometers of the LKO Arosa and PMOD/WRC Davos

Julian Gröbner[1], Herbert Schill[1], Luca Egli[1], and René Stübi[2]

[1]Physikalisch-Meteorologisches Observatorium Davos, World Radiation Center, Davos Dorf, Switzerland
[2]Meteoswiss, Payerne, Switzerland

**Correspondence:** Julian Gröbner (julian.groebner@pmodwrc.ch)

**Abstract.** Total column ozone measured by Brewer and Dobson spectroradiometers at Arosa and Davos, Switzerland, have systematic seasonal variations of around 1.5% using the standard operational data processing. Most of this variability can be attributed to the temperature sensitivity of approx. +0.1%/K of the ozone absorption coefficient of the Dobson spectroradiometer (in this study D101). While the currently used Bass&Paur ozone absorption cross-sections produce inconsistent results for Dobson and Brewer, the use of the ozone absorption cross-sections from Serdyuchenko et al. (2013) in conjunction with an effective ozone temperature dataset produces excellent agreement between the investigated four Brewers (of which two double Brewers), and Dobson D101. Even though other ozone absorption cross-sections available in the literature are able to reduce the seasonal variability, all of those investigated produce systematic biases in total column ozone between Brewer and Dobson of 1.1% to 3.1%. The highest consistency of total column ozone from Brewers and Dobson D101 at Arosa/Davos of 0.1% is obtained by applying the Rayleigh scattering cross-sections from Bodhaine et al. (1999), the ozone absorption cross-sections from Serdyuchenko et al. (2013), the effective ozone temperature from either ozonesondes or ECMWF, and the measured line-spread functions of Brewer and Dobson. The variability between Brewer and Dobson for single measurements of 0.9% can be reduced to less than 0.5% for monthly means and 0.3% on yearly means. As show here, the proposed methodology produces consistent total column ozone datasets between Brewer and Dobson spectroradiometers of better than 1%. For colocated Brewer and Dobson spectroradiometers, as is the case for the Arosa/Davos total column ozone times series, this allows the merging of these two distinct datasets to produce a homogeneous time series of total column ozone measurements. Furthermore, it guarantees the long-term future of this longest total column ozone time-series, by proposing a methodology how to eventually replace the ageing Dobson spectroradiometer with the state-of-the art Brewer spectroradiometer.

## 1 Introduction

The world's longest continuous total column ozone time series was initiated in 1926 at the Lichtklimatisches Observatorium (LKO), at Arosa, in the Swiss Alps. It consists of a suite of six instruments, 3 Dobson and 3 Brewer spectroradiometers, which measure the direct spectral solar irradiance at several narrow wavelength bands between 305 nm and 345 nm. Since 2010, Brewer B072 has been relocated to the Physikalisch-Meteorologisches Observatorium Davos, World Radiation Center





(PMOD/WRC), located in the nearby valley of Davos, at 12 km horizontal distance from Arosa, to investigate the possibility
of relocating the measurements from LKO Arosa to PMOD/WRC Davos (Stübi et al. , 2017). As demonstrated from the simul-
taneous measurements by the Brewer spectroradiometers located at LKO Arosa and PMOD/WRC, Davos, total column ozone
measurements between the sites are consistent to better than 1%, which is well within the uncertainties of the measurements.
Dobson D101 was moved in January 2016, in order to validate the consistency between the two sites using Dobson spectrora-
diometers as well (Stübi et al. , 2020). In 2018, two further instruments, Dobson D051, and Brewer B156, were moved from
LKO Arosa to PMOD/WRC, with the objective of finalising the relocation of the whole LKO instrumentation to PMOD/WRC
by June 2021.

The Brewer spectroradiometers were installed at LKO Arosa in 1988 (Brewer B040), 1991 (Brewer B072), and 1998 (Brewer
B156) to complement the Dobson spectroradiometers already in operation. The objective has been to eventually replace the
Dobson spectroradiometers with the more modern and automated Brewer spectroradiometers. However colocated measure-
ments between Dobson and Brewer spectroradiometers have shown seasonal variations of the order of 2% to 3% which have
been linked to the effective ozone temperature  (Köhler et al. , 2018; Vanicek et al. , 2012; Staehelin et al. , 1998; Redondas et
al. , 2014). These known systematic variabilities between the two instrument types have so far precluded a merging of datasets
from these instruments and have led to the continuous simultaneous operation of the Dobsons and Brewers at LKO Arosa.

A study by Redondas et al.  (2014) on the use of different ozone absorption cross-sections in Brewer and Dobson total column
ozone retrieval algorithms has shown that most of the observed discrepancies can be reconciled by taking into account the tem-
perature dependence of the ozone absorption cross-sections of the Dobson spectroradiometer. The study investigated five ozone
absorption cross-sections, accessible at the ACSO (Absorption Cross Sections of Ozone) initiative web page (http://igaco-
o3.fmi.fi/ACSO). Redondas et al.  (2014) came to the conclusion that the cross-sections measured by Serdyuchenko et al.
(2013) substantially reduced the observed seasonal dependence of the differences between Dobson and Brewer.
In this study, we revisit the issue of the consistency of Brewer and Dobson total column ozone measurements discussed in
Redondas et al.  (2014), with the following additional elements:

– An additional dataset of ozone absorption cross-sections measured in the frame of the EMRP project ATMOZ (M. Weber,
personal communication).

– A dataset of absorption cross sections measured within the ESA SEOM-IAS project (Scientific exploitation of Opera-
tional Missions - Improved Atmospheric Spectroscopy Dabases) (Birk et al. , 2018).

– Recent line spread functions from Dobson D101 measured using the portable tunable radiation source (TuPS) (Šmid et
al. , 2020).

– A comprehensive dataset of total column ozone measurements from 2016 to 2020 for Brewers MKII B040, MKII B072,
MKIII B156, MKIII B163, and Dobson D101.





## 2 Instrumentation, datasets and methodology

### 2.1 Total ozone column measurement

Brewer and Dobson spectroradiometers retrieve the atmospheric total column ozone from the absorption of direct solar radiation by the atmosphere, as described by the Beer-Lambert law,

$$I(\lambda) = I_0(\lambda)e^{-\tau(\lambda)\mu} \tag{1}$$

where $\lambda$ is the wavelength, $I_0$ is the solar irradiance at the top of the atmosphere, $I$ the irradiance measured at the surface, $\tau$ the optical depth of the atmosphere and $\mu$ the slant pass (airmass). The equation can be rewritten as,

$$\tau(\lambda) = \frac{\log I_0(\lambda) - \log I(\lambda)}{\mu} \tag{2}$$

to retrieve the optical depth $\tau$ at a specific wavelength. In the ultraviolet range between 300 nm and 340 nm the main atmospheric absorber is ozone. Even though sulphure dioxide also absorbs in this wavelength range, its amount in the atmosphere above Arosa/Davos is so small that it can be neglected here. The only further attenuation comes from Rayleigh scattering ($\tau_\beta$), which needs to be subtracted from the total optical depth $\tau$ to retrieve the optical depth for ozone,

$$\tau_{O_3}(\lambda) = \alpha(\lambda)X\mu_{O_3} = \frac{\log I_0(\lambda) - \log I(\lambda) - \tau_\beta\mu_\beta}{\mu_{O_3}} \tag{3}$$

where $X$ is the total amount of ozone in the atmospheric column, $\alpha(\lambda)$ the ozone absorption cross-section and $\mu_\beta$ and $\mu_{O_3}$ the effective airmass for Rayleigh scattering and ozone absorption respectively.

While the Brewer spectroradiometer uses a slit mask to measure quasi-simultaneously the solar irradiance at four wavelengths, the Dobson spectroradiometer measures the intensity ratio at two wavelengths pairs by adjusting an optical wedge to attenuate the higher of the two intensities with respect to the other. The total column ozone $X$ is then retrieved by a linear combination of the solar radiation measurements at these four wavelengths,

$$X = \frac{F_0 - F - \Delta\beta\mu_\beta}{\Delta\alpha\mu_{O_3}} \tag{4}$$

according to $F = \sum_{i=1}^{4} w_i \cdot \log I(\lambda_i)$, where $F_0$, $F$, $\Delta\beta$, and $\Delta\alpha$ are the weighted sums of the logarithms of the top of atmosphere solar irradiance, logarithms of the measured solar irradiance, Rayleigh scattering coefficient, and ozone absorption coefficient at wavelength $\lambda_i$ respectively. The weighting coefficients $w_i$ are $w_B^i = [+1 - 0.5 - 2.2 + 1.7]$ and $w_D^i = [+1 + 1 - 1 - 1]$ for the Brewer and Dobson spectroradiometers respectively, with $i$ defined by the instrument wavelength according to Table 1.

### 2.2 Brewer and Dobson spectroradiometers

The spectroradiometers operating at Davos and Arosa belong respectively to the global Brewer and Dobson networks and are traceable to their respective network reference instruments, Dobson D083 via the RVI regional Dobson D064 from Hohenpeissenerg, Germany, and the Brewer Triad maintained by the Regional Brewer calibration center - Europe (RBCC-E), at the



Izaña Atmospheric Research Center, Tenerife, Canary Islands, Spain (León-Luis et al. , 2018). For details on the maintenance and calibration schedules we refer to the publications of Scarnato et al. (2010), and Stübi et al. (2017). The calibration
reports from the latest calibration campaigns which took place in July 2018 are available at the WMO GAW report portal (https://library.wmo.int/index.php?lvl=notice_display&id=21655#.X5rL8IhKhaQ9) and on request from the authors. The consistency between the Brewer spectroradiometers operated at Arosa/Davos is of the order of 0.5% (Stübi et al. , 2017). A recent analysis for the manually and automatically operated Dobson spectroradiometers at Davos and Arosa have found a similar consistency of 0.5% (Stübi et al. , 2020).

The four Brewer spectroradiometers consist of two single monochromator Brewers MKII B040 and B072, and two double monochromator Brewers MKIII B156 and B163. From the three Dobsons, only D101 and D062 are continuously measuring total column ozone, while D051 is usually reserved for Umkehr measurements and is only occasionally fitted with the zenith prism for direct solar radiation measurements. As the line spread function measurements with the TuPS were not successful on D062 due to the low sensitivity of the instrument, only D101 will be used in this study.

The top of the atmosphere calibration constants $F_0$ were obtained through direct comparisons with the reference spectroradiometers from each network during site-visits to Arosa/Davos. Following the standard operating procedures, the measurements are then corrected for instrument drifts between calibrations by applying standard lamp corrections which are automatically made several times per day for Brewer spectroradiometers and manually on a weekly schedule for Dobson spectroradiometers. All four Brewer spectroradiometers were calibrated at Arosa in 2016 and 2018, while Brewer B163 also took part in the RBCC-
E campaigns at El Arenosillo, Spain, in the years in between (2017 and 2019). The default calibration period for Dobsons has been scheduled every four to five years, resulting in calibrations of the Dobson spectroradiometers at LKO Arosa in 2012, 2017 and 2018. The third calibration in 2018 was necessary due to the bad weather conditions in 2017, which did not allow for a reliable calibration at the time.

### 2.2.1 Line spread functions

The spectral resolution and wavelength calibration of the Brewer spectroradiometers is obtained from the measurements of a set of spectral emission lines from several spectral discharge lamps, as described in Gröbner et al. (1998).The ozone absorption coefficients are then retrieved by convolving the selected ozone absorption cross-sections with the spectral line spread functions at the specified wavelength positions. Since the wavelength setting of the ozone measurement position of the Brewer does not coincide with the measurement of the spectral emission lines, the line spread functions, approximated with a trapezoid, are
interpolated between nearby spectral emission line measurements.

The spectral characterisation of the Dobson spectroradiometers at LKO Arosa was not performed in the past due to the need of a tuneable monochromatic source. Instead, nominal ozone absorption coefficients, based on the primary Dobson instrument, D083, were used by all Dobsons of the global Dobson network. To improve the situation, a tuneable portable source was developed in 2016, in the frame of the project , to allow the individual spectral characterisation of Dobson spectroradiometers (Šmid
et al. , 2020). This device was used in 2019 to measure the line spread functions of Dobson D101 and to calculate the ozone absorption coefficients based on these measurements.





**Table 1.** Central wavelength positions of the Brewer B040, B072, B156, B163 spectroradiometers and Dobson D101. The weighting coefficients $w_{\mathrm{B}}^{i}$ and $w_{\mathrm{D}}^{i}$ are also shown.

| Brewer | Wavelength /nm | | | | $w_{\mathrm{B}}^{i}$ | Dobson | Wavelength /nm | Wavelength /nm | $w_{\mathrm{D}}^{i}$ |
|---|---|---|---|---|---|---|---|---|---|
| Slits | B040 | B072 | B156 | B163 | | Slits | D101(TuPS) | Nominal | |
| 3 | 310.085 | 310.065 | 310.048 | 310.053 | 1 | A1 | 305.663 | 305.500 | 1 |
| 4 | 313.533 | 313.517 | 313.507 | 313.507 | -0.5 | D1 | 317.697 | 317.500 | 1 |
| 5 | 316.831 | 316.813 | 316.813 | 316.800 | -2.2 | A2 | 325.306 | 325.000 | -1 |
| 6 | 320.040 | 320.015 | 320.016 | 320.005 | 1.7 | D2 | 340.062 | 339.900 | -1 |

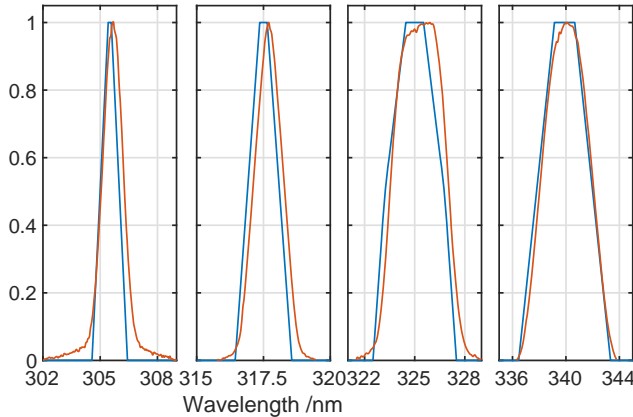

**Figure 1.** Line spread functions of Dobson D101 measured in 2019 using the portable tuneable source (TuPS) (red), and the nominal parametrized slits for the A and D wavelength pairs (blue).

Table 1 shows the spectral characteristics of the four Brewers and of Dobson D101 used in this study, as well as the nominal wavelength positions of the Dobson according to the Dobson handbook. Figure 1 shows the measured line spread functions of Dobson D101 as well as the nominal slit functions used for the calculation of the operational ozone absorption cross-sections according to Bernhard et al. (2005).

### 2.3 Effective ozone temperature and ozone height from ozone sondes

The effective ozone temperature $T_{\mathrm{eff}}$ is calculated from ozone and temperature profiles measured by ozone sondes launched every two to three days at the aerological station of MeteoSwiss in Payerne, 220 km distance from Davos. The effective ozone temperature is obtained from the integral of temperature $T(z)$ and ozone density $O_3(z)$ profiles according to,

$$T_{\mathrm{eff}} = \frac{\int T(z) \cdot O_3(z) dz}{\int O_3(z) dz} \tag{5}$$



**Figure 2.** Effective ozone temperature calculated from ozone sonde launches at Payerne (blue curve), and from the ECMWF reanalysis (yellow curve). A periodic function is fitted to the radiosonde data, resulting in an amplitude of 5.7 K and an average temperature of 225.2 K.

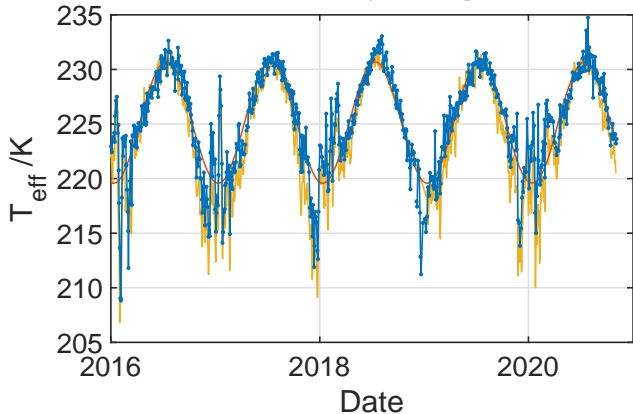

where $z$ represents the vertical height above ground. The ozone and temperature above the sonde burst height, which occurs typically around 30 km is obtained by extending the measured ozone and temperature profiles with the standard ozone and temperatures taken from the standard US atmosphere (NOAA, 1976) and normalised to the measurements at the sonde burst height.

Figure 2 shows the effective ozone temperature over Payerne from 2016 to 2020. The effective ozone temperature shows a distinct seasonal variation, with an average effective ozone temperature of 225.2 K throughout the period and with a 95% coverage between 215 K in winter and 232 K in summer. The average seasonal variability of the effective ozone temperature is 11.4 K as estimated from a periodic fit to the data.

    During the winter, the effective ozone temperature can fluctuate significantly, with changes of up to 15 K within a few days 135   (e.g. from 225 K on 27 January 2016, to 209 K on 3 February 2016 and back to 224 K on 10 February 2016). The effective ozone temperatures on days without ozone sonde launches were obtained from linearly interpolating between available measurements. The resulting dataset was smoothed with a 10 day running average to reduce the day to day noise.

    We have also investigated the effective ozone temperature obtained from the ECMWF reanalysis (http://www.temis.nl/ climate/efftemp/overpass.html) for Payerne, shown as the yellow curve in the figure. The two datasets agree very well at larger 140   temperature above 220 K (deviations below 2 K), while at lower temperatures the temperatures from ECMWF systematically underestimate the effective ozone temperatures obtained from the radio-sondes by up to 4 K. Even though we will concentrate on the dataset obtained from the radio-sondes, we will briefly comment on the use of the ECMWF dataset for the recalculation of the Dobson total column ozone time series later in the manuscript.

    The corresponding effective ozone height $H_{eff}$ was also calculated by replacing the temperature profile in Eq. 5 with the 145   height profile. The resulting average effective ozone layer over Payerne is found at 22.3 km, with a minimum of 20.2 km during the winter and a maximum of 23.8 km during the summer. The impact of this variability on the calculation of the ozone slant path for the measurements at Arosa/Davos is at most 0.3% at an airmass of 3.9 (SZA of 76), which is the maximum





that can be reached due to the mountains blocking the horizon at both sites. Notwithstanding the small expected impact, for consistency reasons we have incorporated the calculated ozone layer height in the calculations of total column ozone from the

Brewers and Dobson in this study. The effective molecular scattering height relevant for Rayleigh scattering was kept at 5 km as used in the nominal operating procedure.

## 2.4  Ozone absorption cross-sections

The operational ozone absorption coefficients in use by the Brewer and Dobson networks are based on the ozone cross-sections from Bass and Paur  (1985) and are presented in  Redondas et al.  (2014) and the references therein. Furthermore,

we have selected datasets of ozone absorption cross-sections available for the wavelength range 300 nm to 345 nm, which cover the wavelength range used by Brewer and Dobson spectroradiometers to retrieve total column ozone. The selected cross-sections are also available at several ozone temperatures, in order to calculate the ozone absorption cross-sections for the actual effective ozone temperature at our sites. If not specified otherwise, these cross-sections were retrieved from the ACSO web-page (http://igaco-o3.fmi.fi/ACSO/cross_sections.html). The following six ozone absorption cross-sections were used in this

study:

BPOp  The nominal ozone absorption coefficients based on the standard operational procedures of the Brewer and Dobson networks. For Brewer spectroradiometers, these are based on the Bass&Paur cross-sections at a temperature of 228 K, while for the Dobson spectroradiometers, they are equal to the ones of Dobson D083 as calculated by Komhyr et al. (1993) at 227 K.

IGQ  The quadratic polynomial temperature approximation of the Bass&Paur ozone absorption cross sections from the IGACO web-page (file bp.par).

DBM  The high resolution dataset of Daumont et al.  (1993), Brion et al.  (1993), and Malicet et al.  (1995) at five temperatures between 218 K and 295 K. Due to the lack of measurements at temperatures below 218 K, the quadratic temperature approximation gives inconsistent values at temperatures lower than 218 K. As a compromise, we have therefore decided

to calculate instead a linear temperature dependence function from the three lowest temperatures of the DBM dataset to extrapolate the DBM cross-sections to temperatures below the minimum temperature of 218 K.

IUP&IUP_A  Two datasets of ozone absorption cross-sections measured by the University of Bremen, IUP in 2013. (Serdyuchenko et al. , 2013), and IUP_ATMOZ in 2017 (M. Weber, personal communication). The dataset IUP was used in previous investigations and has been selected by the WMO as the future new reference cross-sections for the Brewer and Dobson

networks (M. Tully, personal communication). The IUP_ATMOZ cross-sections between 295 nm and 350 nm were measured in 2017 during the project EMRP ATMOZ and have improved noise characteristics in this wavelength region when compared to the original IUP cross-sections. Here, we used the quadratic polynomial temperature approximation provided for both datasets.



**Table 2.** Weighted ozone absorption coefficients of Brewer B040, B072, B156, B163 spectroradiometers and the AD weighted coefficients for Dobson D101 in cm$^{-1}$. The relative deviations to the operational ozone absorption coefficients in the first row are also shown. IUP_A stands for IUP_ATMOZ. The temperature coefficient in %/K of the ozone absorption coefficients are calculated over the temperature range found at Arosa/Davos.

| X-Section | O$_3$ abs. coef. $\Delta\alpha$ in cm$^{-1}$ | | | | Rel. Dev | Temp. Dep. | $\Delta\alpha_{AD}$ in cm$^{-1}$ | | Rel. Dev. | Temp. Dep. |
|---|---|---|---|---|---|---|---|---|---|---|
| | B040 | B072 | B156 | B163 | to BPOp in % | in %/K | D101 | Param. | to BPOp in % | in %/K |
| BPOp | 0.3337 | 0.3398 | 0.3407 | 0.3402 | 0.0 | – | 1.432 | 1.432 | 0.0 | – |
| IGQ | 0.3290 | 0.3352 | 0.3363 | 0.3356 | -1.4 | +0.104 | 1.422 | 1.417 | -0.7/-1.0 | 0.126 |
| IUP | 0.3371 | 0.3429 | 0.3442 | 0.3433 | +0.9 | +0.010 | 1.429 | 1.425 | -0.2/-0.5 | 0.091 |
| IUP_A | 0.3420 | 0.3479 | 0.3492 | 0.3481 | +2.4 | +0.001 | 1.434 | 1.429 | +0.1/-0.2 | 0.113 |
| DBM | 0.3483 | 0.3540 | 0.3554 | 0.3543 | +4.1 | -0.066 | 1.428 | 1.423 | -0.3/-0.6 | 0.091 |
| ACS | 0.3351 | 0.3409 | 0.3425 | 0.3412 | +0.4 | -0.030 | 1.399 | 1.394 | -2.3/-2.6 | 0.042 |

ACS  A recent dataset measured in the frame of the ESA project SEOM-IAS between 243 nm and 346 nm and at 193 K, 213 K, 233 K, 253 K, 273 K, and 293 K (Birk et al. , 2018). We determined a quadratic polynomial temperature dependence function from this dataset to calculate the ozone absorption cross-sections for this study.

### 2.4.1   Ozone absorption coefficients

The ozone absorption coefficient $\alpha_i$ at wavelength $\lambda_i$ is calculated by convolving the selected ozone absorption cross-sections $\sigma(\lambda, T_{eff})$ for a particular effective ozone temperature $T_{eff}$ with the line spread function $S_i(\lambda)$ of the spectroradiometer,

$$\alpha_i = \frac{\int \sigma(\lambda, T_{eff}) \cdot S_i(\lambda) d\lambda}{\int (S_i(\lambda) d\lambda},  \tag{6}$$

where $i$ specifies the central wavelength at which the measurement is taken. The ozone absorption coefficient $\Delta\alpha$ from equation 4 can then be calculated,

$$\Delta\alpha = \sum_{i=1}^{4} w_i \cdot \alpha_i  \tag{7}$$

Table 2 summarises the ozone absorption coefficients $\Delta\alpha$ at a temperature of 228 K ($-45^o$ C) for the ozone cross-sections described in the previous section. For historical reasons, the calculations to retrieve ozone from Dobson and Brewers have been performed using the common logarithm base 10 instead of the natural logarithm, thus $\log$ in the previous equations is replaced with $\log_{10}$, and $\Delta\alpha$ and $\Delta\beta$ are divided accordingly by $\log_{10}$. The table also shows the relative deviations between the different ozone absorption cross-sections relative to the operational coefficients. The total column ozone values calculated from these ozone absorption coefficients directly scale with these values and give a first indication by how much the total ozone values are shifted when using one or the other cross-sections.

The ozone absorption coefficients versus effective ozone temperature for Brewer B156 and Dobson D101 are shown in Fig. 3. Brewers B040, B072, and B163 follow the dependence of Brewer B156 very closely and are not separately shown.

**Figure 3.** Temperature dependence of the ozone absorption coefficients normalised at 228 K for Brewer B156 and Dobson D101 for the five ozone absorption cross-sections IGQ, IUP, IUP_ATMOZ, DBM, and ACS. The ozone absorption coefficients at 228 K are those shown in Table 2.

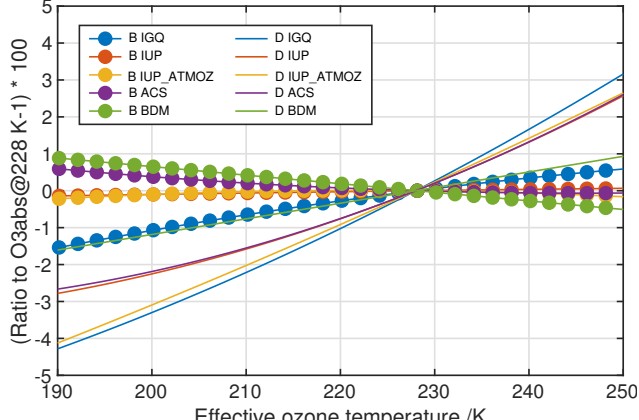

As can be seen in Table 2, the relative differences between the ozone absorption coefficients are the same for all Brewers. This indicates that a change in absorption cross sections for a Brewer spectroradiometer operated according to the standard procedure, produces the same relative change in total column ozone. This is an important result for the Brewer network, because it implies that the extra-terrestrial constants retrieved from a comparison to a reference instrument would not change significantly when the ozone absorption cross-sections would get updated in the future. Redondas et al. (2014) reached a similar conclusion using the characterisations from 123 Brewers in the EUBREWNET database (http://www.eubrewnet.org/).

The same conclusion regarding a change of cross-sections also applies to the Dobson instrument, since the ozone absorption coefficients, calculated either using the measured line spread functions or the parametrized slits, result in the same relative offsets. However as seen in Table 2, the ozone absorption coefficient increases slightly by 0.3% when using the measured slit functions with respect to the parametrized slits. Similar shifts between -0.6% to +0.3% were found for three Dobson spectroradiometers which were characterised during the project EMRP ATMOZ (Köhler et al. , 2018), indicating that the specific spectral characteristics of a Dobson spectroradiometer have an impact of at least this magnitude on the retrieval of total column ozone. This demonstrates the need for an individual spectral characterisation of each Dobson spectroradiometer to reach uncertainties of 1 % or lower.

## 2.5 Rayleigh scattering coefficient

The Rayleigh scattering coefficients $\beta_i$ at wavelength $i$ are calculated by replacing the ozone absorption cross-section $\sigma(\lambda, T_{\mathrm{eff}})$ in Eq. 6 with the Rayleigh scattering coefficient $\beta(\lambda)$, calculated using the parametrization from Bodhaine et al. (1999) and scaled with the station pressure. The Rayleigh scattering coefficient $\Delta\beta$ is then the weighted mean of the individual coefficients according to equation 7.


### 2.5.1 Dobson D101

For Dobson D101, the nominal Rayleigh scattering coefficients used by the operational procedure are 0.114 cm$^{-1}$ and 0.104 cm$^{-1}$ for the A and D wavelength pairs respectively. The coefficients calculated using the parametrized slit functions are

0.1140 cm$^{-1}$ and 0.1043 cm$^{-1}$ for the A and D pairs respectively, and therefore equal to the nominal values to within their stated precision. The coefficients calculated for the actual line spread functions measured with the TuPS and using Bodhaine et al. (1999) are 0.1146 cm$^{-1}$ and 0.1038 cm$^{-1}$ for the A and D wavelength pairs. The resulting total column ozone correction for the AD pair is 7.0 DU $((0.114-0.104)/1.432)$ for the nominal ozone and Rayeigh cross-sections. When using the measured (TuPS based) line spread functions, the total ozone column correction is between 7.5 DU to 7.7 DU $((0.1146-0.1038)/\Delta\alpha_{\mathrm{AD}})$

depending on which ozone absorption cross-section is used. Thus the change in total column ozone resulting from using the Rayleigh scattering coefficients calculated for the actual Dobson D101 wavelengths and using the coefficients from Bodhaine et al. (1999) produces an ozone offset relative to the ozone values retrieved using the operational constants of between -0.4 DU to -0.6 DU for a mean station pressure at Arosa/Davos of 840 mbar and depending on the actual ozone absorption cross-sections used.

### 2.5.2 Brewer

The same calculations were performed for the Brewer spectroradiometers. While using the nominal Rayleigh scattering coefficients $\beta_i = \{0.4620, 0.4410, 0.4220, 0.4040\}$ found in the operational procedure result in a total column ozone correction of +0.3 DU, the use of the Rayleigh scattering coefficients from Bodhaine et al. (1999) produce a total column ozone offset of -2.6 DU, resulting in an offset of -2.9 DU relative to the operational ozone retrieval of the Brewer procedure (For B156,

$\beta_i = \{0.4585, 0.4372, 0.4180, 0.4004\}$, and $\Delta\beta = 0.00098$). For Arosa/Davos at a pressure of 840 mbar, the total column ozone shift resulting from the use of the Rayleigh scattering coefficients from Bodhaine et al. (1999) therefore results in a significant reduction of the total column ozone values of 2.4 DU with respect to the total column ozone calculation using the nominal constants of the Brewer operational procedure.

## 3 Results and discussion

The datasets obtained from each ozone absorption cross-section were compared between the four Brewer spectroradiometers with the corresponding ones of Dobson D101. The comparisons were made by applying consistently the ozone effective temperatures as calculated from the ozone sondes launched at Payerne (see section 2.3), the Rayleigh scattering coefficients (see section 2.5), and the ozone absorption coefficients calculated from the spectral characterisations performed on each instrument. We will primarily discuss the total column ozone comparison between Brewer B156 and Dobson D101, while the results of

the comparisons between the other Brewers to Dobson D101 are summarised in the corresponding tables, and the figures of the comparisons are provided in the supplementary material to this paper. The comparisons are based on simultaneous





**Figure 4.** Left figure: Relative differences of total column ozone between Brewer B156 and Dobson D101 for the period 1 January 2016 to 30 June 2020 using the operational ozone retrieval procedure with the ozone absorption cross-sections from (Bass and Paur , 1985). The yellow curve is a periodic fit to the data, while the black curve is obtained by combining the effective ozone temperature with the linear temperature dependence retrieved from the dataset. Right figure: the same relative differences, shown versus effective ozone temperature.

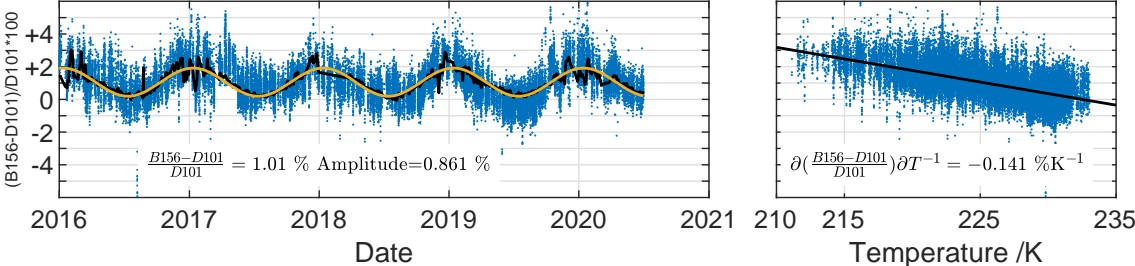

measurements within a 5 minute window, totaling more than 40000 measurements over the time period considered. The solar zenith angle varied between $23.3^o$ and $75.7^o$ and the corresponding ozone airmass between 1.1 and 3.9.

### 3.1 Operational Bass&Paur cross-sections

The left panel of Fig. 4 shows the relative difference between the total column ozone from Brewer B156 to Dobson D101, calculated using the operational procedures. The amplitude of a sinusoidal fit to the data with a period of one year gives an amplitude of 0.86%, with an average offset between the two datasets of 1.0%. If the full variability of the data is examined, then the 95% variability between Brewer B156 and Dobson D101 extends over 3.9% (from -0.75% to +3.2%), which is a significant inconsistency that needs to be addressed. Similar offsets and seasonal variations relative to Dobson D101 are also obtained for

Brewers B040, B072 and B163.

As expected, this seasonal variability correlates very well with the effective ozone temperature as shown in the right panel of Fig. 4. Indeed, the temperature coefficient of -0.14 %/K retrieved from a linear fit to the data, combined with the amplitude of the average seasonal effective temperature variation of 5.7 K, gives an amplitude of 0.80%, in excellent agreement with the amplitude of 0.86% of the periodic fit to the dataset shown in the left panel of Fig. 4. This indicates that the seasonal variability

observed between the Brewers and Dobson D101 is strongly correlated to the effective ozone temperature.

The relative differences between the datasets calculated with different ozone absorption cross sections and taking into account the effective ozone temperature are shown in Fig. 5 with respect to the operational datasets for Brewer B156 and Dobson D101. The panels in the figure are based on the values shown in Table 2 and are therefore mainly a visual illustration of the impact of changing to a different ozone absorption cross-section with its specific ozone temperature dependence. Note that the

Rayleigh correction of -2.4 DU was also included in the operational ozone dataset for Brewer B156, so it cancels out when taking the ratios shown in Fig. 5.

As expected, the amplitude of the periodic sine-fit shown in the sub-figures agree very well with the temperature coefficients calculated for each ozone absorption cross-section shown in Table 2 multiplied with the effective ozone temperature and





also to the seasonal fit of the ozone ratios between Brewer and Dobson shown in black in the figures. This excellent agreement

indicates that the observed seasonal variability is mainly due to the temperature dependence of the ozone absorption coefficients

calculated for Brewer or Dobson.

The most obvious feature apparent in Fig. 5 is the pronounced seasonal variability of the Dobson D101 datasets due to

the inclusion of the effective ozone temperature in the calculations. On the other hand, the average offsets to the operational

dataset are for the most part below 1%, with the exception of the ACS data, with an offset of +2.4%. In contrast, the changes

for Brewer B156 show a much more reduced seasonal variability relative to the operational dataset, which indicates that the

Brewer ozone measurement procedure (wavelength settings and relative weighting) produces an ozone absorption coefficient

largely insensitive to the ozone temperature. On the other hand, the offsets between the ozone absorption coefficients calculated

from these cross-sections vary between +1.7% for the IGQ dataset to -4.1% for DBM and are therefore much larger than for

the Dobson. This larger variability is probably due to the smaller spectral widths of the line spread functions of the Brewer,

which is therefore more sensitive to spectral noise in the cross-sections than from the wider Dobson line spread functions (see

Eq. 6).

## 3.2 Consistency between Brewer and Dobson total column ozone datasets

The comparisons between the datasets of Brewer B156 and Dobson D101 are shown in Fig. 6 for the five cross-sections. The

comparisons between the other three Brewers to D101 are summarised in Table 3 and are shown as figures in the supplement.

A possibe confounding factor could be the different locations of the Brewer and Dobson spectroradiometers during the study

period, with Brewer B156 colocated with Brewer B040 from 2016 to 2018 at Arosa, and then moved to Davos for the remaining

period. However, as discussed in (Stübi et al. , 2017), the observed differences in total column ozone between the two sites has

shown no significant differences, and so no distinction has been made between the instruments located at Arosa or Davos.

As seen in Fig. 6 for Brewer B156 to Dobson D101, the relative differences in total column ozone show substantial variations,

depending on which cross-section is used to calculate total column ozone. The largest seasonal variation in total column

ozone measured by Brewer B156 relative to Dobson D101 is seen with the IGQ cross-sections, with an amplitude of 0.75%

and an average offset of +1.8%. The observed seasonal variation is mainly due to a left-over dependence on effective ozone

temperature as seen by the good agreement between the black and yellow curves. This implies that the temperature dependence

of the IGQ cross-sections fail to account for the actual temperature dependence of the total column ozone retrieved by either

Brewer or Dobson.

As can be seen in Table 3, there is a slight systematic difference in the seasonal variability of the single Brewer spec-

troradiometers relative to Dobson D101, compared to the one seen with the double Brewer spectroradiometers. Since single

Brewer spectroradiometers are affected by stray-light at large ozone slant path, we have applied a correction based on the

stray-light correction determined during the RBCC-E campaign in 2018, with respect to the reference double Brewer B185.

The corrections are small, but slightly improve the consistency with the results from the double Brewers, as can be seen in

Table 3.







**Figure 5.** Relative differences in total column ozone calculated with different ozone absorption cross-sections and effective ozone temperature relative to the operational ozone datasets for Brewer B156 (left panels) and Dobson D101 (right panels). The red curve in the figure represents a sinusoidal fit to the data with a period of 1 year. The average offset and amplitude of the sine wave are also shown.

The seasonal variabilities between the Brewers and Dobson D101 are significantly improved when the cross-sections from ACS, DBM, IUP, or IUP_ATMOZ are used. The results from the individual Brewers are highly consistent, with a remaining





**Figure 6.** Left panels: total column ozone relative differences between Brewer B156 to Dobson D101 for the five investigated cross-sections for the period 1 January 2016 to 30 June 2020. The black line represents the impact of the linear temperature coefficient calculated from this data using the effective ozone temperature, as shown on the corresponding panels on the right. The yellow curves represent a sine function fit to the data with a period of one year. The average offset and amplitude of the fitted sine curve are shown in the panels. Right panels: the same data shown with respect to the effective ozone temperature. The black line is a linear fit, and the value of the gradient is shown in the panels.

seasonal amplitude equal or less than 0.2%, (only B156 using IUP has a slightly higher amplitude of 0.4%). In contrast, large

differences are observed in the offsets between the total column ozone calculated from these cross-sections for the Brewers and Dobson D101. The largest discrepancies of -3.2% and -1.7% are observed with the DBM and ACS cross-sections respectively,





**Table 3.** Average offset and seasonal amplitude in % of the relative differences between Brewers B040, B072, B156, and B163 to Dobson D101 for the five investigated ozone absorption cross-sections. These results are extracted from the appropriate figures, such as Fig. 6 for Brewer B156. The last row in the table (Br avg) represents the average of the results of the four Brewers relative to D101.

| Brewer /D101 | Op Offset | Op Amp. | IGQ Offset | IGQ Amp. | DBM Offset | DBM Amp. | ACS Offset | ACS Amp. | IUP_A Offset | IUP_A Amp. | IUP Offset | IUP Amp. |
|---|---|---|---|---|---|---|---|---|---|---|---|---|
| B040 | +0.7 | 0.56 | +1.5 | 0.45 | -3.6 | 0.03 | -2.1 | 0.09 | -1.6 | 0.06 | -0.4 | 0.12 |
| B040* | +0.9 | 0.72 | +1.7 | 0.62 | -3.5 | 0.10 | -2.0 | 0.06 | -1.5 | 0.08 | -0.3 | 0.20 |
| B072 | +0.8 | 0.31 | +1.6 | 0.19 | -3.5 | 0.28 | -2.0 | 0.34 | -1.5 | 0.31 | -0.3 | 0.13 |
| B072* | +1.3 | 0.68 | +2.1 | 0.58 | -3.1 | 0.05 | -1.6 | 0.01 | -1.1 | 0.04 | -0.1 | 0.05 |
| B156 | +1.0 | 0.86 | +1.8 | 0.75 | -3.4 | 0.24 | -1.9 | 0.19 | -1.4 | 0.22 | -0.2 | 0.41 |
| B163 | +1.7 | 0.66 | +2.4 | 0.54 | -2.7 | 0.05 | -1.2 | 0.01 | -0.7 | 0.01 | +0.5 | 0.19 |
| Br avg* | +1.2 | 0.73 | +2.0 | 0.62 | -3.2 | 0.11 | -1.7 | 0.07 | -1.2 | 0.09 | 0.0 | 0.21 |

\* Stray-light correction applied to the single Brewers, based on the calibration relative to the double Brewer B185 obtained during the RBCC-E calibration campaign in 2018.

while the best agreement of 0.0% is seen with IUP, and -1.2% with IUP_ATMOZ. These results confirm the findings by Redondas et al. (2014), which also saw similar large discrepancies with DBM, and the best consistency with the IUP cross-sections.

The new results with ACS and IUP_ATMOZ complement the previous study by Redondas et al. (2014), and support the fact that the Dobson spectroradiometers are sensitive to effective ozone temperature and need a correction based on the temperature coefficients calculated from the ozone absorption cross-sections.

### 3.2.1  Total column ozone using the effective ozone temperature from ECMWF

The availability of a global dataset of effective ozone temperature becomes crucial if the decision is made to reprocess the whole
historical dataset of the global Dobson network. Such a global dataset is available through the afore-mentioned ECMWF effective ozone temperature climatology, which is available daily since 1956 on a global scale. Without entering in a detailed discussion on this topic, we have evaluated how the re-processed Dobson total column ozone dataset using either the ozonesonde or ECMWF-based effective ozone temperature dataset agrees at Arosa/Davos. As mentioned previously, the values of effective ozone temperatures from ECMWF are systematically biased low, the differences increasing with decreasing temperatures by
up to 4 K. Nevertheless, the total column ozone dataset reprocessed using the ECMWF temperatures is only 0.1% lower on average than the ozonesonde based one, and the standard deviation of 0.2% between the two datasets is also extremely low. Therefore at least for the northern mid-latitude site Arosa/Davos, either effective ozone temperature dataset can be used for the recalculation of the total column ozone from Dobson D101 with a bias of at most 0.1% and a resulting uncertainty of less than 0.2%.





**Figure 7.** Relative difference in total column ozone between Brewer B156 and Dobson D101 (red points) and B163 versus D101 (blue points) with respect to ozone slant pass. The data was restricted to conditions with effective ozone temperature between $225 \pm 2$ K. The error bars represent the standard deviation of the datapoints at selected airmasses $\pm$ 50.

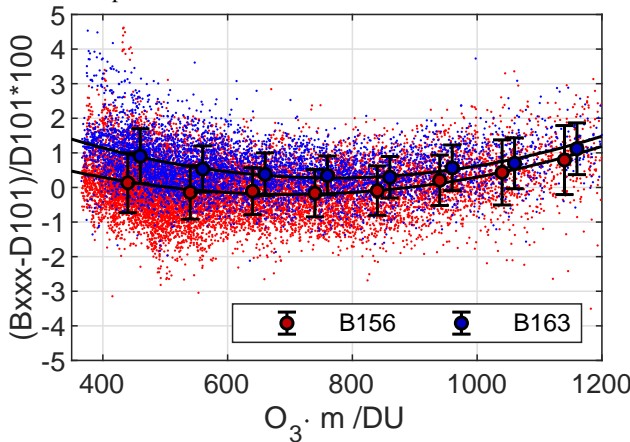

### 3.2.2 Dependence on ozone slant pass

A possible influence of stray light on the total column measurements was investigated by correlating the relative differences between the double Brewers B156 and B163 to Dobson D101 against ozone slant pass. The data was filtered by using only effective ozone temperatures between $225 \pm 2$ K. As can be seen in Fig. 7, the relative differences vary between about $\pm 1\%$, with a slight curvature, indicating some small unexplained features between the instruments, which could be due to the actual extra-terrestrial constants coming from the independent calibrations performed on Brewers and Dobson. However the instruments do not deviate systematically at larger ozone slant pass, which would be an indication for a stray-light impact on the ozone retrieval. The same comparison using the two uncorrected single Brewers B040 and B072 (not shown), shows the impact of stray-light due to the single monochromator Brewers, which under-estimate ozone at large airmass due to the stray light coming from longer wavelengths.

## 4 Conclusions

The highest consistency between Brewer and Dobsons is obtained with the IUP ozone absorption cross-sections (Serdyuchenko et al. , 2013) and applying an effective ozone temperature correction. The effective ozone coefficient of the Brewer spectro-radiometer, as already noted by Redondas et al. (2014), is only weakly sensitive to a change in effective ozone temperature (0.1% per 10 K, see Table 2) , while for the Dobson, the sensitivity to effective ozone temperature is considerably larger, with 0.9% per 10 K change.

Based on the results of the double Brewers B156 and B163, the lowest residual seasonal variability between Brewer and Dobson is obtained with the ACS or IUP_ATMOZ cross-sections, indicating that their temperature dependence are the most



consistent of the investigated ozone absorption cross-section datasets. Unfortunately, the observed offsets of more than 1% indicate that there is some spectral inconsistency in the datasets which precludes their use as common ozone absorption cross-

sections for the Brewer and Dobson networks.

Due to this significant sensitivity to effective ozone temperature, the total column ozone calculation procedure for the Dobson spectroradiometer requires this additional ancillary information to produce total column ozone corrected for the seasonal variability observed at LKO Arosa and at other stations. This ancillary information could be obtained from ozone sonde launches as done in this study, or from climatological reanalyses such as from the ECMWF effective ozone temperature

reanalysis (http://www.temis.nl/climate/efftemp/overpass.html).

As shown here, the use of measured spectral line spread functions to calculate the ozone absorption coefficient of the Dobson D101 improves the agreement with the colocated Brewer spectroradiometers. This well established procedure for Brewer spectroradiometers would also improve the general agreement within the Dobson network, since even the reference Dobsons used for the calibration of network Dobson instruments show relative differences in ozone absorption coefficients of

up to 0.6% when the actually measured line spread functions are used instead of the nominal ones (Köhler et al. , 2018).

## 4.1 Implications for the total column ozone series of LKO Arosa/Davos

The results of this study are based on individual measurements by well maintained Brewers and Dobson spectroradiometers within the period 2016 to 2020 covering nearly 5 years of continuous measurements at airmasses between 1.1 and 3.9.

Following the decision made by the WMO Scientific Advisory Group on Ozone and UV (WMO SAG O3UV), the total

column ozone time series of LKO Arosa/Davos, which are based on a series of Dobson spectroradiometers, will need to be reprocessed using the IUP ozone absorption cross-sections and applying a correction for the effective ozone temperature following the methodology outlined here. Similarly, the total column ozone measurements from the Brewer spectroradiometers operating at LKO Arosa/Davos, need to be reprocessed, using the updated Rayleigh scattering cross-sections and the IUP ozone absorption cross-sections.

Once these datasets are reprocessed, both datasets from Brewer and Dobson can be merged to form a homogenised total column ozone dataset for LKO Arosa/Davos. The resulting uncertainty of the merged dataset can be estimated by the comparison of total column ozone measurements obtained between the colocated Brewer and Dobson instruments during the investigated period. As shown in the bottom left panel of Fig. 6 and in the figures in the supplement, the comparison of coincident single measurements within a 5 minute window between Brewers B040, B072, B156, and B163 to Dobson D101 gives an excellent

agreement with offsets smaller than 0.5% and standard deviations of 0.9% or smaller. For daily means, this variability decreases to 0.7%, for monthly means to 0.45%, and for the yearly means over the investigated period the relative difference between the reprocessed datasets of Brewers and Dobson D101 reaches 0.3%.



*Author contributions.* JG analysed the datasets and wrote the manuscript, HS processed the datasets, LE analysed the line-spread functions of Dobson D101 and calculated the ozone absorption coefficients, and RS is the responsible scientist for the Arosa/Davos total column ozone times series.

*Competing interests.* We declare no competing interests.

*Acknowledgements.* The study was partly funded by the ESA project QA4EO, No. QA4EO/SER/SUB/09 and by the project INFO3RS funded by MeteoSwiss, Grant number 123001926.





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
