# Peer review of "Consistency of total column ozone measurements between the Brewer and Dobson spectroradiometers of the LKO Arosa and PMOD/WRC Dayos"

_Atmospheric Measurement Techniques, 2020_

## Author Comment (AC1)

Dear James Kerr,

Thank you for your valuable comments and suggestion to add two publications as reference to our manuscript. The calculated temperature sensitivities of Brewer ozone absorption coefficients in Kerr, 2002 correspond very well to our calculations as pointed out in your comment. We will also add in our revised manuscript the fact that the very low temperature sensitivity of the Brewer ozone absorption coefficients you observed are consistent with the IUP and IUP_A findings, giving even more confidence in these cross-sections.

---

## Author Response (AR1)

We would like to thank the reviewers for their valuable comments and suggestions. The lines describing the changes are according to the revised manuscript.

Reviewer #1:

*21: Rephrase because as it written now it implies that all instruments (including Brewers) were installed in 1926!*
We will do so, and also add a reference describing the history of the Arosa ozone measurements.

Line 21-22: The world's longest continuous total column ozone time series was initiated in 1926 at the Lichtklimatisches Observatorium (LKO), at Arosa, in the Swiss Alps with Dobson D002 (Staehelin et al. , 2018).

*24: In addition to horizontal, state the vertical displacement of the two stations.*

The LKO Arosa is located at an altitude of 1850 m and PMOD/WRC is located at 1590 m elevation, giving a difference in elevation of 260 m.

Line 25-26: located in the nearby valley of Davos (1590 m.a.s.l.), at 12 km horizontal distance from Arosa (1850 m.a.s.l.),

*27: In the abstract it is stated that there is a seasonal variability of 1.5%, while here that there a consistency of within 1%. Which of the two is more accurate? Furthermore, in line 87 this number is further reduced to 0.5%.*
The numbers might have been misleading. The 1.5% seasonal variability is seen between colocated Brewer and Dobson spectroradiometers, and has been documented already in 1988 by Kerr et al (reference added to the revised manuscript). The consistency within the Brewers and within the Dobsons is much higher, as documented by the publications by Stübi et al. We have verified the numbers and have found them as accurate so we have left the manuscript unchanged.

*64: Aerosols and NO2 also absorb in this range. Although for Davos and Arosa their effect should be negligible, these species should be mentioned for completeness.*
As the reviewer pointed out, the influence of aerosols and NO2 to the ozone retrieval can be neglected at Davos and Arosa. Nevertheless, we will mention them in the manuscript. We have also added a reference to aerosol optical measurements at Davos which confirm this assumption.
Line 66-69: Even though sulphure dioxide and nitrogen dioxide also absorb in this wavelength range, their amount in the atmosphere above Arosa/Davos is so small that it can be neglected here. The influence of the scattering and absorption by atmospheric aerosols on the ozone retrieval is minimized by using ratios of measurements at close-by wavelengths, as discussed later.
Line 84-87: The weighting coefficients for the Brewer were specifically selected to cancel any absorption that is linear across the four wavelengths, which is a good approximation for the aerosol optical depth in this wavelength range. It should be noted that apart from very rare aerosol intrusions, the aerosol optical depth at Arosa/Davos is very close to background levels and has therefore a negligible influence on the ozone retrieval (Nyeki et al. , 2012).

*68: Actually, α(λ) is the absorption coefficient and not the cross section*
Replaced accordingly, see line 73.

*109: This sentence is unclear for non-experts, please rephrase: "…does not coincide with the emission lines of the spectral lamps, the line…"*
Lines 126-130: First paragraph of section 2.2.2 was changed: The spectral resolution and wavelength calibration of the Brewer spectroradiometers is obtained from the measurements of a set of spectral emission lines from several spectral discharge lamps, as described in Gröbner et al. (1998).The width of the line spread functions at the ozone position of the Brewer spectroradiometer, approximated with a trapezoid, are obtained by linearly interpolating between nearby spectral emission line measurements. The

ozone absorption coefficients are then retrieved by convolving the selected ozone absorption cross-sections with these calculated line spread functions.

*212: The procedure for determining the error in the total ozone due to the use of different Rayleigh cross sections could be slightly expanded so that inexperienced readers can follow it better. Alternatively, a reference could be provided to improve understanding.*

The influence of the Rayleigh coefficients on the ozone retrieval is explicit in equation 4 on page 3. We will refer to this equation in the Section 2.5 on the Rayleigh coefficient calculation and expand it as suggested by the reviewer.

Line 238: As shown explicitly in Eq. 4, Rayleigh scattering can cause a bias in the total ozone retrieval if it is not accounted properly.

*238: However, if new Rayleigh cross sections are used, then the calibration of the instrument would change so this offset of about 2.4 DU would be finally compensated.*

The Brewer network currently uses Rayleigh coefficients with unknown origin. When Rayleigh coefficients based on known Rayleigh cross-sections are calculated, an offset of 2.4 DU occur in the ozone, affecting the whole Brewer network. Clearly, this cancels out when Brewers are compared with one another, but when comparing to independent instruments such as Dobson or Satellites this needs to be taken into account. No changes were made to the manuscript.

*248: The term "ozone airmass" cannot be understood by non-expert readers. Line 69 defines it as "effective airmass for ozone absorption" so the term could also be used here.*

Line 273: and the corresponding effective airmass for ozone between 1.1 and 3.9.

*315: ECMWF does not provide the effective temperature but the temperature profiles from which the effective temperature can be calculated. Moreover the ozone profiles that are needed for the calculation of Teff are available from other sources which should be mentioned.*

We agree with the comment of the reviewer concerning our use of the ECMWF effective temperatures. We will add a reference describing how this data-set is produced and update the web-site link where this data can be downloaded.

Lines 159:161: The effective temperature data is produced from a combination of MSR reanalysis and real-time data. For real-time data, the algorithm uses temperature profiles from ECMWF operational data while for past data, ECMWF re-analysis data is used (van der A et al. , 2010).

*253: Please explain what is meant by 95% variability. Does it refer to the 95% of the data?*

Lines 277-278: The relative difference in total column ozone between Brewer B156 and Dobson D101 extends over 4.0% (from -0.69% to +3.28% defined as the 95% data coverage),

*280: In addition to noise in the cross sections another reason could be the different wavelengths used in Brewers compared to Dobsons in conjunctions with the spectral variability of the cross sections.*

Lines 306-307: Another reason could be from the different wavelengths used in Brewers compared to Dobsons in conjunction with the spectral variability of the ozone absorption cross sections.

307: The average deviation for IUP_ATMOS is the same (though of opposite sign) with the Operational settings. However, the spread becomes smaller amplitude 0.09 vs 0.75 and this should be mentioned.
We have revised the corresponding paragraph:

Lines 329-335: The seasonal variabilities between the Brewers and Dobson D101 are significantly improved when any of the cross-sections ACS, DBM, IUP, or IUP_ATMOZ are used instead of the operational cross-sections or IGQ. The results from the individual Brewers are highly consistent, with a remaining seasonal amplitude equal or less than 0.2%. In contrast, large differences are observed in the offsets between the total column ozone calculated from these cross-sections for the Brewers and Dobson D101. The largest discrepancies of -3.2% and -1.7% are observed with the DBM and ACS cross-sections respectively, while the best agreement of 0.0% is seen with IUP, and -1.1% with IUP_ATMOZ. These results confirm the findings by Redondas et al. (2014), which also saw similar large discrepancies with DBM, and the best consistency with the IUP cross-sections.

*343-345: I think this sentence is somewhat misleading. It is not clear what is meant by "precludes their use as common ozone absorption cross-sections". If I understand correctly, the same cross-section can be used in both types of instruments as long as their temperature sensitivity is taken into account for each instrument.*

As suggested by the reviewer, we will reformulate our statement in the conclusion on the inconsistency of some cross-sections when used by the Brewer and Dobson algorithms to make it better understandable. We wanted to make the statement that some cross-sections, like for example DBM, produce a significant offset in the ozone derived by Brewer and Dobson respectively, which indicates that these cross-sections cannot be used when trying to homogenise the measurements of these two networks.

Line 369: Relative differences in total column ozone of 0.3% or less are observed between all four Brewers and Dobson D101 when the IUP cross-sections are used.

*332: Since a supplement already exists, I suggest to include this figure in the supplement, to demonstrate the difference of the stray-light effect of the single Brewers.*

The stray-light effect of the single brewers has been extensively documented during Brewer comparisons performed during previous RBCC-E campaigns. As suggested by reviewer #2, we will add DOI links to these reports which contain the required information for the single brewers as compared to the double Brewers. We will also modify the manuscript accordingly to refer to these reports. We will also add a figure to the supplement as suggested by the reviewer.

Line 358-359: The same comparison using the two single Brewers B040 and B072 is shown in Figure 4S in the supplement.

*335: The conclusions section starts very abruptly. Please start by at least identifying the instrument's location.*

Line 360: Total column ozone measurements from Brewer and Dobson spectroradiometers located at Davos/Arosa in the Swiss Alps have been analysed.

All technical comments were also addressed in the revised manuscript.
* * *
Reviewer #2:

*Most of the datasets used are not available and will difficult the reproducibility of the results, this include, ozone cross-section of the EMPR-ATMOZ, line Spread functions, and the Brewer /Dobson Ozone datasets.*

The Brewer and Dobson measurements will eventually be submitted to the World Ozone and UV database of the WMO. As this could take some time, we have collected and submitted the relevant

data used in this publication to an open-access repository and added the DOI to the manuscript to provide access for interested scientists: https://doi.org/10.5281/zenodo.4559802

Line 397: Data availability. The datasets used in this study can be downloaded from https://doi.org/10.5281/zenodo.4559802 (10.5281/zenodo.4559802)

*Straylight correction is applied to the single brewer, this should be detailed in the methodology section. Also, the straylight on Dobson can be explained Can be estimated by TUPS measurements?*

We will add the information on the stray-light correction of the single Brewers in the revised manuscript. The TuPS measurements only provide information of the line spread function over about one order of magnitude and only over a narrow spectral range, as shown in Figure 1. Unfortunately, this does not allow to estimate the stray-light contribution.

Lines 112-124: We have added section 2.2.1 to the revised manuscript:

2.2.1 Stray light correction of single Brewers
The total column ozone measurements from the single Brewer spectroradiometers B040 and B072 were corrected for stray light according to the methodology described in Redondas et al. (2019). The empirical correction function was obtained by direct comparison with the reference double monochromator Brewer B185 of the Regional Brewer Calibration Center Europe during the campaign in Arosa in July 2018. The total column ozone corrected for stray light is obtained by correcting the extraterrestrial constant $F_0$ of Eq. 4,
$F_{sr0} = F_0 + a(osc)_b$ (5)
where osc represents the ozone slant path, and a and b are empirical constants derived for B040 and B072 with respect to the reference Brewer B185. The constants a and b are shown in Figure 18 (a = ⊡10:2, b = 5:21) and Figure 27 (a = ⊡23:8, b = 3:92) of the report (Redondas et al. , 2019) for B040 and B072 respectively. The corrections in total column ozone increase exponentially with increasing ozone slant path and are quite small; for example at an ozone slant path of 1000 the corrections for B040 and B072 are +0.3% and +0.6% respectively. The largest corrections of +1.1% and +1.6% occur at an ozone slant path of 1390 DU for B040 and B072 respectively.

*Section 2.1 Total ozone measurements: Aerosol term and its cancelation is missing from the discussion.*

As already mentioned by reviewer #1, we will mention the aerosol and NO2 terms in section 2.1. Both can be considered negligible for Arosa and Davos due to their altitudes of 1800 m and 1590 m respectively as well as their location in the mountains, far distant from urban pollution.

Lines 84-87: The weighting coefficients for the Brewer were specifically selected to cancel any absorption that is linear across the four wavelengths, which is a good approximation for the aerosol optical depth in this wavelength range. It should be noted that apart from very rare aerosol intrusions, the aerosol optical depth at Arosa/Davos is very close to background levels and has therefore a negligible influence on the ozone retrieval (Nyeki et al. , 2012).

line 100: Calibration reports can be cited by his DOI in particular
Reference has been added: Lines 444-446: A. Redondas, S.F. León-Luís, J. López-Solano, A. Berjón, V. Carreño, Thirteenth Intercomparison Campaign of the Regional Brewer Calibration Center Europe, Joint publication of State Meteorological Agency (AEMET), Madrid, Spain and World Meteorological Organization (WMO), Geneva, Switzerland, WMO/GAW Report No. 246, 2019, https://dx.doi.org/10.31978/666-20-018-3.

*line 130: Please explain the normalization of the ozone sounding.*

We will clarify the ozone sounding normalisation in the paper. It is done by using the ozone sounding readings of ozone and temperature one kilometer below its burst altitude, and extending the ozone and temperature profiles using the US standard atmosphere to extend both profiles to 100 km, normalised to the readings at the selected altitude.

Lines 146-149: The ozone and temperature above the sonde burst height, which occurs typically around 30 km is obtained by extending the measured ozone and temperature profiles with the standard ozone and temperatures taken from the standard US atmosphere (NOAA, 1976) which are normalised to the sonde temperature and ozone density values just below the sonde burst height.

*line 138: please correct the link ( the correct one ends in .php)*

We will correct the link to the updated location of the ECMWF effective ozone temperature datasets and add also the following reference, which describes how this dataset is obtained:

van der A, R. J., Allaart, M. A. F., and Eskes, H. J.: Multi sensor reanalysis of total ozone, Atmos. Chem. Phys., 10, 11277–11294, https://doi.org/10.5194/acp-10-11277-2010, 2010.

*line 150: Brewer and Dobson use different ozone effective heights on the operational procedure for the air mass calculation the effect of the ozone height is different, even if the effect is reduced due to the horizon minor please clarify.*

For our calculations we have used the same constant ozone layer height of 22 km for the Brewer and Dobson ozone calculations. We will adapt the text in the manuscript accordingly.

Lines 170-172: As we are primarily interested in the comparison of Brewer and Dobson total column ozone, we have decided to use a constant ozone layer height of 22 km in the calculations of total column ozone from the Brewers and Dobson in this study.

line 230: Units missing for Rayleigh coefficients.
Units were added. See lines 256-263.

line 275: Explanation for the large difference on the offset of ACS dataset.
The ozone retrieved by Dobson with the ACS dataset shows large discrepancies when compared to the ozone retrieved by other cross-sections investigated in this manuscript (see Figure 5). Similar features can be seen with the DBM dataset when applied to the Brewer. We will modify the text in the last paragraph of Section 3.1 to clarify this point.
Lines 329-335: The seasonal variabilities between the Brewers and Dobson D101 are significantly improved when any of the cross-sections ACS, DBM, IUP, or IUP_ATMOZ are used instead of the operational cross-sections or IGQ. The results from the individual Brewers are highly consistent, with a remaining seasonal amplitude equal or less than 0.2%. In contrast, large differences are observed in the offsets between the total column ozone calculated from these cross-sections for the Brewers and Dobson D101. The largest discrepancies of -3.2% and -1.7% are observed with the DBM and ACS cross-sections respectively, while the best agreement of 0.0% is seen with IUP, and -1.1% with IUP_ATMOZ. These results confirm the findings by Redondas et al. (2014), which also saw similar large discrepancies with DBM, and the best consistency with the IUP cross-sections.

*line 325: To explicit the straylight, could usefully use a common calibration for both instruments, Brewer is calibrated against the Dobson or vice versa, using low OSC conditions and then see the comparison at high OSC conditions. We have to take into account that the Dobson has a considerably bigger FOV (Dobson nominal from FOV 7º to 8 º whereas the Brewer is around 2º-3º) and is more affected by atmospheric straylight.*

The final suggestion concerning the possibility of straylight coming from the larger field of view of the Dobson spectroradiometer is interesting. Even though the effect is probably quite small at Arosa and Davos due to the low aerosol optical depth, it could still cause a bias at large ozone slant path which could be misinterpreted as coming from the spectral stray light of the monochromator.

The contribution of forward scattered radiation in a 10° field of view, compared to a field of view of the Brewer was investigated in Gröbner & Kerr, 2001, and was found to be very small at the high altitude location of Mauna Loa, Hawaii (Gröbner, J., and J. Kerr, Ground-based determination of the spectral ultraviolet extraterrestrial solar irradiance: Providing a link between space-based and ground-based solar UV measurements, J. Geophys. Res., 106, 7211-7217, 2001.).

We will discuss this possibility in our revised manuscript. We will consider how to address this issue in a future experiment.

Line 355: or from the forward scattered radiation into the larger field of view of the Dobson ($8_o$ versus $1{:}5_o$ for Dobson and Brewer respectively).

Figure 5: BOp and Bop are confusing terms of the first panel, please change.

The figure was updated accordingly.